# Incidence and etiology of chronic pulmonary infections in patients with idiopathic pulmonary fibrosis

**Kyuto Odashima**[1]*, **Naho Kagiyama**[1], **Tetsu Kanauchi**[2], **Takashi Ishiguro**[1], **Noboru Takayanagi**[1]

1 Department of Respiratory Medicine, Saitama Cardiovascular and Respiratory Center, Saitama, Japan,
2 Department of Radiology, Saitama Cardiovascular and Respiratory Center, Saitama, Japan

* k-odashima@jikei.ac.jp

**Data Availability Statement:** All relevant data are within the manuscript and figures. All these files are available from the protocols.io database

## Abstract

### Background

The incidence and etiologies of chronic pulmonary infection (CPI) in patients with idiopathic pulmonary fibrosis (IPF) have been poorly investigated.

### Methods

We conducted a retrospective study of 659 patients with IPF to assess the incidence, etiologies, and risk factors of CPI development. CPI was defined if the etiology of infection was diagnosed one or more months after the onset of symptoms or upon the appearance of new shadows on pulmonary radiological images.

### Results

At IPF diagnosis, 36 (5.5%) patients had CPI, and 46 (7.0%) patients without CPI at IPF diagnosis developed CPI over a median follow-up period of 6.1 years. The incidence density of CPI development was 18.90 cases per 1000 person-years. Detected organisms from these 46 patients were *Mycobacterium avium* complex in 20 patients, other nontuberculous mycobacteria in 4, *M. tuberculosis* in 7, *Aspergillus* spp. in 22, and *Nocardia* sp. in one. In a multivariate Cox regression hazard model, PaO$_2$ <70 Torr and KL-6 $\geq$2000 U/mL were associated with CPI development.

### Conclusions

Nontuberculous mycobacteria, *M. tuberculosis*, and *Aspergillus* and *Nocardia* spp. were the four most frequent etiologies of CPI in patients with IPF. During follow-up of IPF, clinicians should pay attention to the development of CPI, especially in patients with PaO$_2$ <70 Torr or KL-6 $\geq$2000 U/mL.

(accession number dx.doi.org/10.17504/protocols.
io.bb2piqdn.).

**Funding:** The authors received no specific funding
for this work.

**Competing interests:** The authors have declared
that no competing interests exist.

## Introduction

Idiopathic pulmonary fibrosis (IPF) is a specific form of chronic, progressive fibrosing interstitial pneumonia of unknown cause. It occurs primarily in older adults, is limited to the lungs, and is associated with the histopathologic and/or radiologic pattern of usual interstitial pneumonia [1, 2]. IPF is characterized by progressive worsening of dyspnea and lung function and is associated with a poor prognosis [1, 2]. Comorbidities including pulmonary hypertension, chronic obstructive pulmonary disease (COPD), lung cancer, pulmonary embolism, and pulmonary infections can occur in IPF [1–4]. The identification and prompt treatment of comorbidities may have a clinically significant and meaningful effect on overall outcome for patients with IPF [4].

A pooled analysis of outcomes in the control arms of several large randomized, placebo-controlled trials searching for effective treatments represents a unique opportunity to evaluate the natural history of IPF. In studies reporting infective events, the standardized rate of lower respiratory tract infections (LRTIs) was 172.5 events per 1000 patient-years, whereas the overall standardized rate of pneumonia was 45.6 events per 1000 patient-years [5]. LRTIs were more common in trials allowing low-dose prednisolone use compared with those that did not (227.1 vs 63.4 infections per 1000 patient-years; incident rate ratio 2.56–5.13, p<0.0001) and were less frequent in trials excluding patients with severe disease (153.9 vs 257.8 infections per 1000 patient-years; incident rate ratio 0.45–0.81, p = 0.0003). The authors concluded that further research is needed to look specifically at rates of respiratory infections and possible risk factors for LRTIs and pneumonia in IPF patients.

Not only acute LRTIs and pneumonia but also chronic pulmonary infections (CPI), including pulmonary tuberculosis (TB), nontuberculous mycobacteriosis (NTM), chronic pulmonary aspergillosis (CPA), and nocardiosis, have been reported in patients with IPF [6–16]. Chung et al [7] reported that an atypical manifestation of TB is common in patients with IPF, which may mimic lung cancer or bacterial pneumonia. Hwang et al [9] reported that the NTM in patients with IPF is characterized as lobar/segmental consolidation with or without cavities, different than that in immunocompetent patients without IPF, and can mimic other diseases, especially bacterial or fungal infection. Fibla et al [17] reported that of 296 patients undergoing surgical lung biopsy for interstitial lung biopsy, the organism cultured in 4 patients (*Nocardia* and *Histoplasma* in one each and *Aspergillus fumigatus* in two) resulted in a change in clinical management. We thus thought that the etiologies of CPI, the prevalence of CPI positivity at IPF diagnosis, the incidence density of CPI development during follow-up, and the risk factors of CPI development should be elucidated on a larger scale.

## Methods

### Subjects

From January 1995 through July 2010, 895 patients with IPF were treated at our institution (a 94-bed tertiary referral center for non-tuberculous respiratory disease with 51 beds for TB and 10 intensive care unit beds). Of these patients, 236 were excluded: at IPF diagnosis, 17 had acute exacerbation of IPF, 210 had simultaneous lung cancer, and 9 had been diagnosed as having microscopic polyangiitis. Thus, the study cohort comprised 659 patients who were further divided into four groups according to the presence or absence of CPI at IPF diagnosis and whether CPI did or did not occur during follow-up (Fig 1). All patients fulfilled the criteria for IPF of the American Thoracic Society and European Respiratory Society [2] or the official ATS/ERS/JRS/ALAT statement on IPF [1]. This study was approved by the institutional review board of Saitama Cardiovascular and Respiratory Center (approval no. 2015023). The ethics

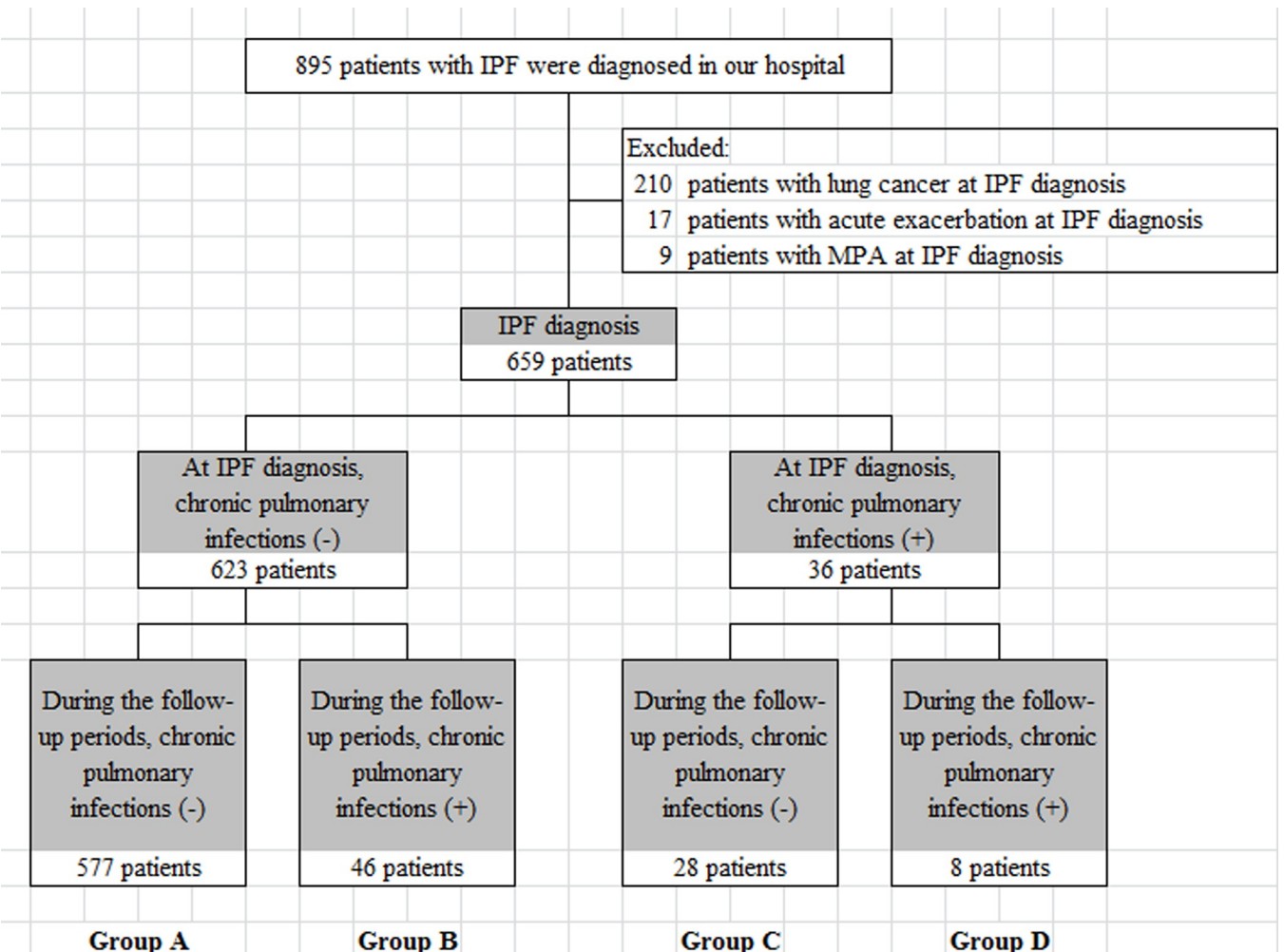

**Fig 1. Diagram of the patients with or without CPI at IPF diagnosis or the follow-up period.** MPA = microscopic polyangiitis.

committee of the same hospital approved the verbal consent procedure. We used an opt-out method and disclosed this to the hospitalized patients. No patient refused to participate.

## Definition of CPI

CPI was defined if the etiology of infection was diagnosed one or more months after the onset of symptoms or upon the appearance of new radiological shadows on chest radiograph or computed tomography (CT). If patients fulfilled the diagnostic criteria for mycobacterial tuberculous lung diseases of the 2015 Japanese Society for Tuberculosis of Japanese guidelines [18], TB was diagnosed when *Mycobacterium tuberculosis* was cultured from respiratory specimens. If patients fulfilled the diagnostic criteria for nontuberculous mycobacterial lung diseases of the 2007 American Thoracic Society/Infectious Disease Society of America guidelines [19], they were diagnosed as having NTM. According to the 2014 Committee guidelines of deep mycosis [20], CPA was diagnosed if patients fulfilled these diagnostic criteria: 1) compatible symptoms continuing >1 month; 2) new consolidation or mass with/without cavity formation, or new consolidation with preexisting cavity expansion; and 3) positive isolation of *Aspergillus* species from respiratory samples or positive serum antibodies to *Aspergillus*

(complement fixation test: Meridian Bioscience Inc., Cincinnati, OH; Bio-Rad Laboratories, Inc., Hercules, CA). Referring to a previous paper [16], nocardiosis was diagnosed when *Nocardia* species was cultured from respiratory specimens along with compatible symptoms and new consolidation.

## Study design

This was a retrospective cohort study. Clinical, laboratory, radiographic, microbiological data, and outcome were collected from medical records. Baseline clinical parameters were obtained within one month of initial diagnosis. If these data were not obtained within this period, we considered them to be unknown. Survival status was obtained from medical records and/or telephone interviews.

## Statistical analysis

Categorical baseline characteristics are summarized by frequency and percent, and continuous characteristics are reported as the mean ± SD. CPI diagnosis was estimated by Kaplan-Meier analysis. Survival was evaluated using a Kaplan-Meier curve and compared between groups using log-lank tests. Cox regression analysis was used to determine whether the following factors at IPF diagnosis increased the risk of CPI: sex, age, body mass index, smoking history, emphysema, diabetes, cerebrovascular disorder, heart disease, chronic hepatic disease, chronic renal disease, gastrointestinal surgery, forced vital capacity (FVC), forced expiratory volume in 1 second (FEV1)/FVC ratio, lung diffusion capacity for carbon monoxide (DLCO), partial pressure of oxygen in arterial blood ($PaO_2$), white blood cell count, erythrocyte sedimentation rate (ESR), C-reactive protein, serum albumin, and Krebs von den Lungen-6 (KL-6). Emphysema was considered present if areas of low attenuation were present on high-resolution CT images. A p-value of <0.05 was considered to be statistically significant. We conducted all statistical analyses with SAS version 9.2 (SAS Institute, Cary, NC).

## Results

### Patient characteristics with or without CPI

Of the 659 patients; 577 (87.5%) were CPI negative at IPF diagnosis and did not develop CPI during follow-up (Group A); 46 (7.0%) were CPI negative at IPF diagnosis but developed CPI during follow-up (Group B); 28 (4.2%) were CPI positive at IPF diagnosis, but did not develop another CPI during follow-up (Group C); and 8 (1.2%) were CPI positive at IPF diagnosis and developed another CPI during follow-up (Group D). Corticosteroids or an immunosuppressant was respectively administered in 133 (23.1%) and 32 (5.5%) of the 577 patients in group A, in 17 (37.0%) and 0 (0%) of the 46 patients in group B, in 4 (14.3%) and 2 (7.1%) of the 28 patients in group C, and in 3 (37.5%) and 2 (25.0%) of the 8 patients in group D (Table 1).

### Etiology and incidence of CPI

At IPF diagnosis, 36 (5.5%) patients had CPI (Groups C+D). Detected organisms were *M. tuberculosis* in 16 (44.4%) patients, *M. avium* complex in 12 (33.3%), *Aspergillus* spp. in 5 (13.9%), *Nocardia* sp. in 2 (5.6%), and *M. avium* complex and *Aspergillus* mixed infection in one (2.8%). In the 623 patients without CPI at IPF diagnosis (Groups A+B), 46 patients (Group B) developed CPI over a median follow-up period of 6.1 years (range, 0.01–15.20 years). Thus, the incidence density of CPI development was 18.90 cases per 1000 person-years (Fig 2). Of these 46 patients, 10 patients (TB in 4, CPA in 4, and NTM in 2 patients) developed CPI after starting corticosteroid therapy, but 36 patients developed CPI without previous

**Table 1. Baseline characteristics of the study patients with IPF according to presence or absence of chronic pulmonary infections at diagnosis or during follow-up.**

| Characteristic | Group A At IPF diagnosis (-) Follow-up (-) | Group B At IPF diagnosis (-) Follow-up (+) | Group C At IPF diagnosis (+) Follow-up (-) | Group D At IPF diagnosis (+) Follow-up (+) |
|---|---|---|---|---|
| No. of patients | 577 | 46 | 28 | 8 |
| Male | 431 (74.7%) | 37 (80.4%) | 22 (78.6%) | 7 (87.5%) |
| Age, years | 69.7 ± 8.3 | 66.0 ± 8.4 | 73.2 ± 10.3 | 72.0 ± 7.8 |
| Smoker | 435 (75.4%) | 38 (82.6%) | 18 (64.3%) | 6 (75.0%) |
| Comorbidity | | | | |
| Emphysema | 158 (27.4%) | 18 (39.1%) | 6 (21.4%) | 4 (50.0%) |
| Diabetes | 91 (15.8%) | 9 (19.6%) | 7 (25.0%) | 3 (37.5%) |
| Cerebrovascular disorder | 39 (6.8%) | 6 (13.0%) | 2 (7.1%) | 0 (0.0%) |
| Heart disease | 108 (18.7%) | 9 (19.6%) | 5 (17.9%) | 2 (25.0%) |
| Chronic hepatic disease | 20 (3.5%) | 2 (4.3%) | 7 (25.0%) | 1 (12.5%) |
| Chronic renal disease | 4 (0.7%) | 0 (0.0%) | 0 (0.0%) | 0 (0.0%) |
| Gastrointestinal surgery | 35 (6.1%) | 4 (8.7%) | 4 (14.3%) | 0 (0.0%) |
| Malignant tumor | 12 (2.1%) | 0 (0.0%) | 1 (3.6%) | 1 (12.5%) |
| Therapy at diagnosis or during follow-up | | | | |
| Steroid | 133 (23.1%) | 17 (37.0%) | 4 (14.3%) | 3 (37.5%) |
| Immunosuppressant | 32 (5.5%) | 0 (0.0%) | 2 (7.1%) | 2 (25.0%) |
| %FVC, % | 75.5 ± 20.5 | 78.9 ± 21.2 | 66.7 ± 14 | 96.6 ± 51 |
| FEV1/FVC, % | 80.9 ± 10.4 | 76.4 ± 14.6 | 84.9 ± 8.2 | 84.9 ± 12.4 |
| %DLCO, % | 77.2 ± 24.2 | 72 ± 18.4 | 50.2 ± 13.8 | 59.1 |
| $PaO_2$, Torr | 72.7 ± 15.1 | 72.1 ± 16.4 | 70.4 ± 16.2 | 78.4 ± 10.6 |
| WBC, /mm$^3$ | 7311 ± 2301 | 7378 ± 2193 | 8126 ± 4457 | 7129 ± 1978 |
| ESR, mm/h | 39.8 ± 29.5 | 43.3 ± 29.5 | 68.5 ± 33.1 | 68.8 ± 39.8 |
| Albumin, g/dL | 4 ± 0.5 | 3.9 ± 0.4 | 3.6 ± 0.6 | 3.8 ± 0.4 |
| CRP, mg/dL | 1.3 ± 3.6 | 1.2 ± 2.2 | 3 ± 4.8 | 4.3 ± 5 |
| KL-6, U/mL | 1056 ± 923.1 | 1084 ± 904.2 | 832.6 ± 630.2 | 677.6 ± 211.8 |

IPF = idiopathic pulmonary fibrosis; FVC = forced vital capacity; FEV1/FVC = forced expiratory volume in 1 second/FVC ratio; DLCO = diffusing capacity for carbon monoxide; $PaO_2$ = partial pressure of oxygen in arterial blood; WBC = white blood cell; ESR = erythrocyte sedimentation rate; CRP = C-reactive protein; KL-6 = Krebs von den Lungen-6.

corticosteroid or immunosuppressant therapy (however, 7 patients received corticosteroid therapy after CPI development). Of these 46 group B patients, 43 (93.5%) had a single pathogen and 3 (6.5%) had two pathogens. Moreover, 5 patients developed second infections (Table 2). Detected organisms were *M. avium* complex in 17 patients, *Aspergillus* spp. in 16, *M. tuberculosis* in 6, and *M. fortuitum*, *M. gordonae*, *M. kansasii*, *M. terrae*, *M. avium* complex and *Aspergillus* mixed infection, *M. avium* complex and *M. tuberculosis* mixed infection, *and M. avium* complex and *Nocardia* mixed infection in one patient each. Thus, the incidence density of NTM, *M. avium* complex (MAC) lung disease, TB, and CPA development was 9.71, 8.04, 2.76, and 8.81 cases per 1000 person-years, respectively. Overall, of 82 patients with CPI, 17 (20.7%) had mixed infection concurrently or sequentially.

## Risk factors of CPI development

In a multivariate Cox regression hazard model, $PaO_2$ <70 Torr was associated with CPI development (adjusted hazard ratio 2.549, 95% confidence interval [CI] 1.107–5.868, p = 0.028), as was KL-6 ≥2000 U/mL (adjusted hazard ratio 3.046, 95% CI 1.065–8.710, p = 0.038) (Table 3).

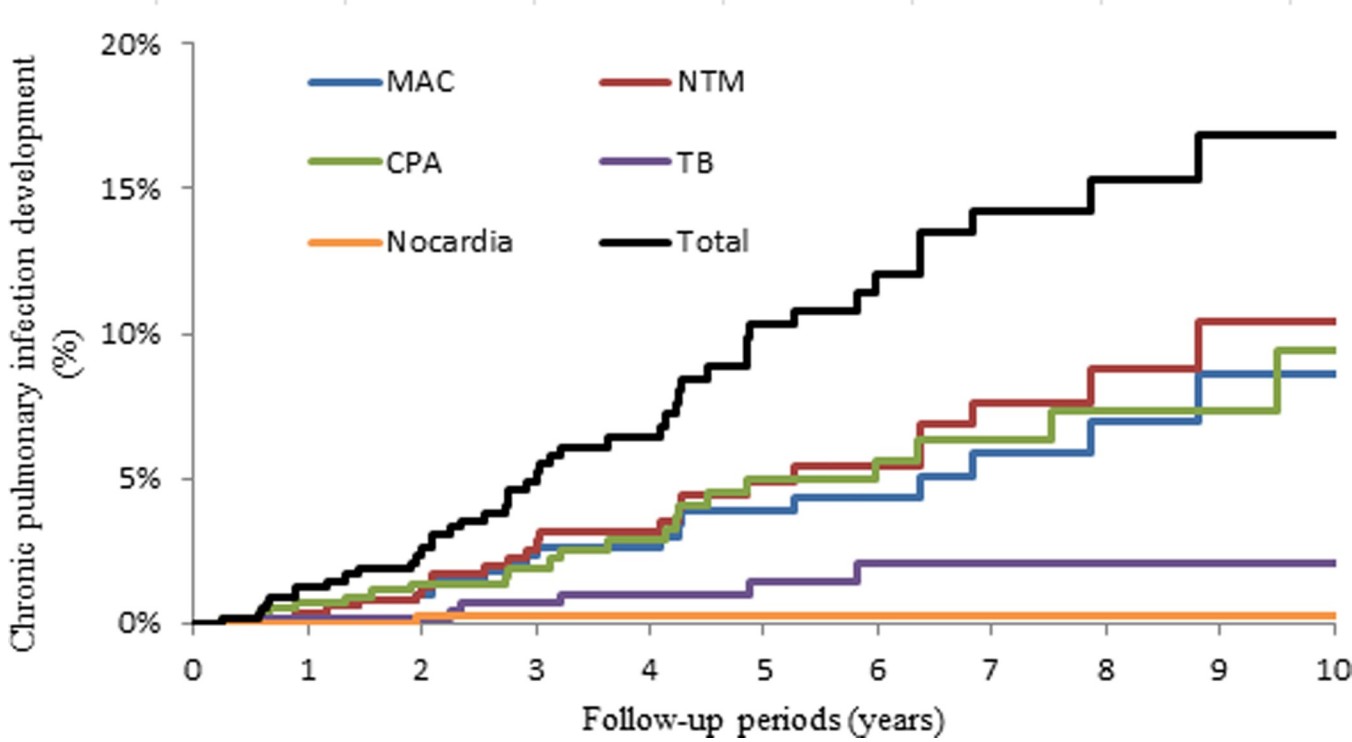

**Fig 2. Kaplan-Meier curves for the time until the development of CPI (Groups A and B).** CPA = chronic pulmonary aspergillosis; MAC = *Mycobacterium avium* complex pulmonary disease; NTM = nontuberculous mycobacteriosis; TB = tuberculosis.

## Mortality in patients with and without CPI at IPF diagnosis (Groups A+B)

Five- and ten-year mortality rates of the patients with and without CPI development were 43.4% and 33.0%, and 68.9% and 72.8%, respectively. Median survival times of the patients with and without CPI development were 6.1 and 6.3 years, respectively. The log-rank test showed no significant difference between the survival curves of the patients with and without CPI development (p = 0.416) (Fig 3). Thus, the presence or absence of CPI did not affect mortality.

## Mortality in patients after CPI onset

Five- and ten-year mortality rates after CPI onset were 57.9% and 70.9%. Median survival time was 4.0 years after CPI onset (Fig 4).

## Causes of death

Of the 659 patients with IPF, death from any cause occurred in 298 (45.2%). Patients without CPI died from progression of IPF (38.6%), acute exacerbation of IPF (16.9%), lung cancer (13.4%), pneumonia (8.7%), pneumothorax (1.6%), *Pneumocystis* pneumonia (1.2%), microscopic polyangiitis (0.8%), non-pulmonary diseases (10.2%), and unknown causes (9.3%). Patients with CPI died from progression of IPF (22.7%), acute exacerbation of IPF (22.7%), lung cancer (6.8%), pneumonia (11.4%), pneumothorax (4.5%), CPA (6.8%), TB (2.3%), and non-pulmonary diseases (22.7%) (Table 4).

## Discussion

This long-term longitudinal study of a large cohort of patients with IPF resulted in five important findings. First, the incidence density of CPI development in patients with IPF but without

**Table 2. Chronic pulmonary infections in patients with idiopathic pulmonary fibrosis (IPF) at IPF diagnosis and during follow-up.**

| No. of chronic pulmonary infections | Types of chronic pulmonary infections | Group B At IPF diagnosis (-) Follow-up (+) | Group C At IPF diagnosis (+) Follow-up (-) | Group D At IPF diagnosis (+) Follow-up (+) |
|---|---|---|---|---|
| No. of patients | | 46 | 28 | 8 |
| First infection | No. of patients | 46 (100.0%) | 28 (100.0%) | 8 (100.0%) |
| | One pathogen | 43 (93.5%) | 27 (96.4%) | 8 (100.0%) |
| | NTM (MAC) | 17 (37.0%) | 11 (39.3%) | 1 (12.5%) |
| | CPA | 16 (34.8%) | 3 (10.7%) | 2 (25.0%) |
| | TB | 6 (13.0%) | 13 (46.4%) | 3 (37.5%) |
| | NTM (*M. fortuitum*) | 1 (2.2%) | | |
| | NTM (*M. gordonae*) | 1 (2.2%) | | |
| | NTM (*M. kansasii*) | 1 (2.2%) | | |
| | NTM (*M. terrae*) | 1 (2.2%) | | |
| | Nocardiosis | | | 2 (25.0%) |
| | Two pathogens | 3 (6.5%) | 1 (3.6%) | |
| | CPA + NTM (MAC) | 1 (2.2%) | 1 (3.6%) | |
| | NTM (MAC) + TB | 1 (2.2%) | | |
| | NTM (MAC) + Nocardiosis | 1 (2.2%) | | |
| Second infection | No. of patients | 5 (100.0%) | | 8 (100.0%) |
| | One pathogen | 5 (100.0%) | | 6 (75.0%) |
| | CPA | 5 (100.0%) | | 1 (12.5%) |
| | NTM (MAC) | | | 4 (50.0%) |
| | TB | | | 1 (12.5%) |
| | Two pathogens | | | 2 (25.0%) |
| | NTM (MAC) + CPA | | | 1 (12.5%) |
| | Nocardiosis + NTM (*M. gordonae*) | | | 1 (12.5%) |

NTM = nontuberculous mycobacteriosis; MAC = *Mycobacterium avium* complex; CPA = chronic pulmonary aspergillosis; TB = tuberculosis.

CPI at IPF diagnosis was 18.90 cases per 1000 person-years. Second, etiologies of CPI were *M. tuberculosis*, NTM, and *Aspergillus* and *Nocardia* species. Third, the incidence density of NTM lung disease, MAC lung disease, TB, and CPA development was 9.71, 8.04, 2.76, and 8.81 cases per 1000 person-years, respectively. Fourth, $PaO_2$ <70 Torr and KL-6 ≥2000 U/mL were risk factors for CPI development. Fifth, 20.7% of patients had CPI caused by concurrent or sequential mixed infection.

In studies reporting infective events, the overall standardized rate of pneumonia was 45.6 events per 1000 patient-years [5], whereas the incidence density of CPI development in our patients with IPF but without CPI at IPF diagnosis was 18.90 cases per 1000 person-years. Because the incidence of CPI in patients with IPF was not so low compared with that of pneumonia, clinicians should pay attention to CPI development during follow-up.

In studies in the 1970–90s that examined patients with idiopathic chronic interstitial lung disease, the positive culture rate for TB was 5%–6.2% [6, 7]. This incidence was 4–5 times higher than that of the general population in the same country at the same time. Park et al [8] reviewed 795 patients with IPF of whom 499 (62.8%) had been treated with immunosuppressants. TB and NTM were found in 35 (4.4%) and 16 (2.0%) of these 795 patients, respectively. Among the 35 patients with TB, 18 (51.4%) were diagnosed at the initial presentation of IPF, and 17 developed TB during follow-up, of whom 13 developed TB after immunosuppressive therapy. Among the 16 patients with NTM, 11 (68.7%) were diagnosed at initial presentation

**Table 3. Univariate and multivariate analysis of the risk of chronic pulmonary infections (Groups A, B).**

| Variable | Univariate Cox regression | | | Multivariate Cox regression Final model | | |
|---|---|---|---|---|---|---|
| | Crude HR | 95% CI | P value | Adjusted HR | 95% CI | P value |
| Age, years | | | | | | |
| <65 | Reference | – | – | | | |
| ≥65, <75 | 0.678 | 0.352–1.305 | 0.244 | | | |
| ≥75 | 0.83 | 0.381–1.809 | 0.639 | | | |
| Sex | | | | | | |
| Female | Reference | – | – | | | |
| Male | 1.171 | 0.564–2.432 | 0.671 | | | |
| BMI (kg/m$^2$) | | | | | | |
| ≥25 | Reference | – | – | Reference | – | – |
| ≥23, <25 | 1.419 | 0.611–3.279 | 0.416 | 1.725 | 0.734–4.056 | 0.211 |
| ≥21, <23 | 1.54 | 0.654–3.629 | 0.323 | 1.772 | 0.741–4.237 | 0.198 |
| <21 | 2.144 | 0.922–4.986 | 0.076 | 2.194 | 0.928–5.187 | 0.073 |
| Unknown | 0.888 | 0.113–6.958 | 0.91 | 1.01 | 0.126–8.135 | 0.992 |
| Smoking status | | | | | | |
| Never smoker | Reference | – | – | | | |
| Ex/current smoker | 1.026 | 0.476–2.210 | 0.947 | | | |
| Unknown | 0 | 0 | 0.992 | | | |
| Emphysema | | | | | | |
| None | Reference | – | – | | | |
| Some | 1.237 | 0.683–2.242 | 0.483 | | | |
| Diabetes | | | | | | |
| None | Reference | – | – | | | |
| Some | 1.274 | 0.615–2.641 | 0.515 | | | |
| Cerebrovascular disorder | | | | | | |
| None | Reference | – | – | | | |
| Some | 2.105 | 0.889–4.983 | 0.09 | | | |
| Heart disease | | | | | | |
| None | Reference | – | – | | | |
| Some | 0.815 | 0.391–1.700 | 0.585 | | | |
| Chronic hepatic disease | | | | | | |
| None | Reference | – | – | | | |
| Some | 1.247 | 0.301–5.159 | 0.761 | | | |
| Chronic renal disease | | | | | | |
| None | Reference | – | – | | | |
| Some | 0 | 0 | 0.989 | | | |
| Gastrointestinal surgery | | | | | | |
| None | Reference | – | – | | | |
| Some | 1.268 | 0.453–3.545 | 0.651 | | | |
| %FVC, % | | | | | | |
| ≥60 | Reference | – | – | | | |
| <60 | 1.313 | 0.452–3.820 | 0.617 | | | |
| Unknown | 1.625 | 0.871–3.033 | 0.127 | | | |
| FEV1/FVC, % | | | | | | |
| ≥70 | Reference | – | – | | | |
| <70 | 1.244 | 0.532–2.905 | 0.614 | | | |
| Unknown | 1.573 | 0.823–3.008 | 0.171 | | | |

*(Continued)*

**Table 3.** (Continued)

| Variable | Univariate Cox regression | | | Multivariate Cox regression Final model | | |
|---|---|---|---|---|---|---|
| | Crude HR | 95% CI | P value | Adjusted HR | 95% CI | P value |
| %DLCO, % | | | | | | |
| ≥70 | Reference | – | – | | | |
| <70 | 1.772 | 0.764–4.109 | 0.182 | | | |
| Unknown | 1.638 | 0.819–3.278 | 0.163 | | | |
| PaO$_2$, Torr | | | | | | |
| ≥70 | Reference | – | – | Reference | – | – |
| <70 | 2.53 | 1.124–5.697 | 0.025 | 2.548 | 1.107–5.868 | 0.028 |
| Unknown | 0.705 | 0.362–1.370 | 0.302 | 0.722 | 0.353–1.478 | 0.373 |
| WBC, /mm$^3$ | | | | | | |
| <10000 | Reference | – | – | | | |
| ≥10000 | 1.443 | 0.566–3.684 | 0.443 | | | |
| Unknown | 1.734 | 0.669–4.490 | 0.257 | | | |
| ESR, mm/h | | | | | | |
| <20 | Reference | – | – | Reference | – | – |
| ≥20, <40 | 2.73 | 1.019–7.312 | 0.046 | 2.66 | 0.980–7.218 | 0.055 |
| ≥40 | 3.349 | 1.271–8.830 | 0.015 | 2.475 | 0.898–6.824 | 0.08 |
| Unknown | 2.379 | 0.909–6.221 | 0.077 | 2.567 | 0.961–6.860 | 0.06 |
| CRP, mg/dL | | | | | | |
| <1.0 | Reference | – | – | | | |
| ≥1.0 | 0.852 | 0.375–1.937 | 0.703 | | | |
| Unknown | 1.423 | 0.645–3.136 | 0.382 | | | |
| Albumin, g/dL | | | | | | |
| ≥3.5 | Reference | – | – | | | |
| <3.5 | 3.024 | 1.021–8.954 | 0.046 | | | |
| Unknown | 1.518 | 0.826–2.789 | 0.179 | | | |
| KL-6, U/mL | | | | | | |
| <1000 | Reference | – | – | Reference | – | – |
| ≥1000, <2000 | 0.927 | 0.313–2.748 | 0.891 | 0.871 | 0.290–2.614 | 0.805 |
| ≥2000 | 3.85 | 1.416–10.469 | 0.008 | 3.046 | 1.065–8.710 | 0.038 |
| Unknown | 1.052 | 0.538–2.058 | 0.882 | 1.03 | 0.517–2.050 | 0.934 |

HR = hazard ratio; CI = confidence interval; FVC = forced vital capacity; FEV1/FVC = forced expiratory volume in 1 second/FVC ratio; DLCO = diffusing capacity for carbon monoxide; PaO$_2$ = partial pressure of oxygen in arterial blood; WBC = white blood cell; ESR = erythrocyte sedimentation rate; CRP = C-reactive protein; KL-6 = Krebs von den Lungen-6.

and 5 during follow-up, and 3 of these 5 were being treated with immunosuppressants. Thus, nearly equal numbers of patients with TB were diagnosed at IPF diagnosis and during follow-up. Moreover, TB developed 3 times more frequently than NTM during follow-up. Among our 23 patients with TB, 16 (69.6%) were diagnosed at initial presentation of IPF, and 7 (30.4%) developed TB during follow-up. Among our 37 patients with NTM, 13 (35.1%) were diagnosed at initial presentation, and 24 (64.9%) developed NTM during follow-up. Thus, TB was diagnosed 3 times more frequently at IPF diagnosis than during follow-up, and NTM developed 3 times more frequently than TB during follow-up. Two reasons for these differences might be that 1) the incidence of TB in the Republic of Korea was nearly 3.7 times higher

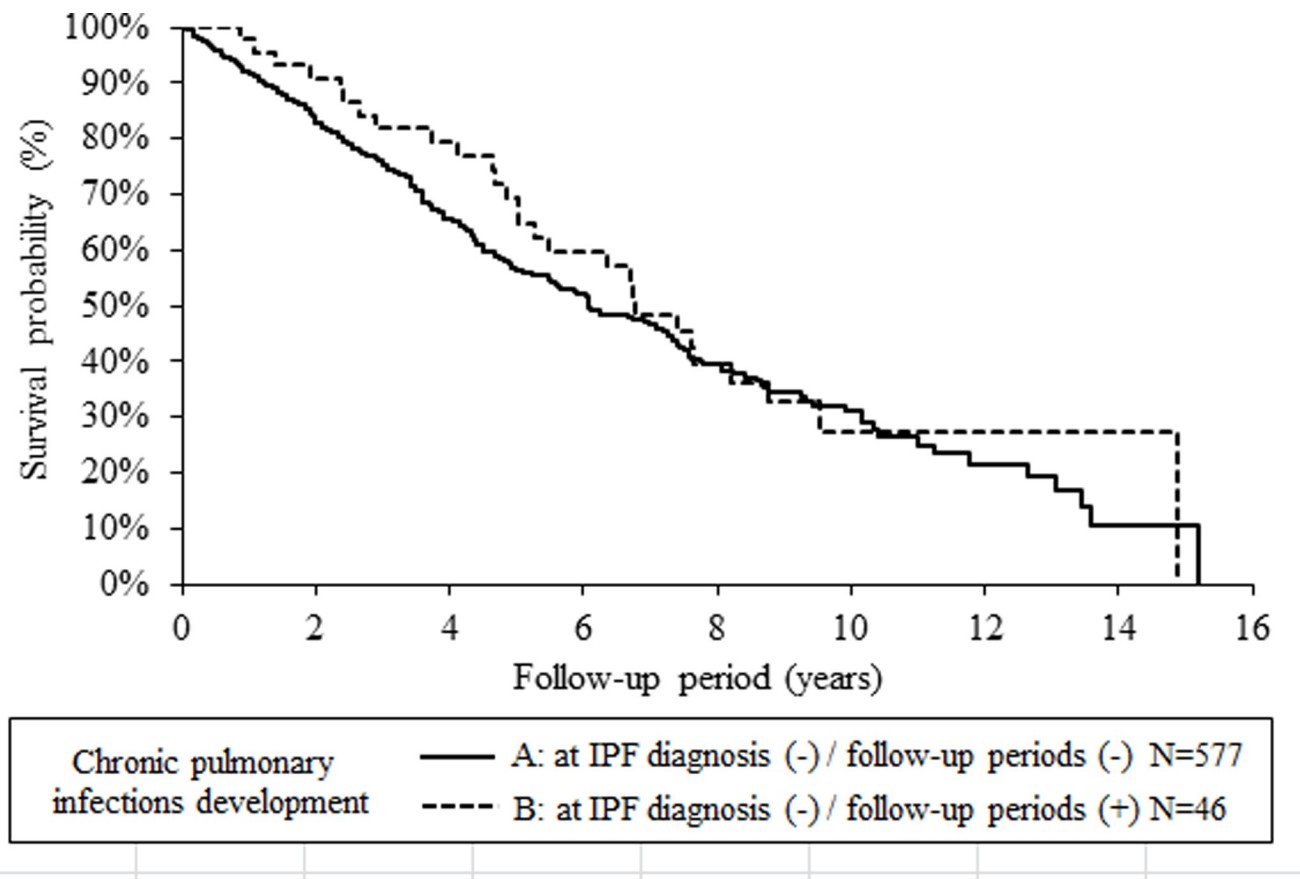

**Fig 3. Kaplan-Meier survival curves according to the presence or absence of CPI (Group A versus B).** A log-rank test showed that the difference was not significant (p = 0.416).

than that in Japan (93 vs 25 per 100,000 person-years in 2005, respectively) [21], and 2) immunosuppressant therapy was 2.6 times more frequently administered in the Park et al study than in the present study (62.8% vs 23.8%, respectively). The most frequently detected organism in our 36 patients with CPI at IPF diagnosis was *M. tuberculosis* (44.4%), but during follow-up, of the 46 patients with CPI, *M. tuberculosis* was detected in only 13.0%. Because our hospital has 51 beds for TB and TB patients tend to be referred to our hospital, it would be reasonable for *M. tuberculosis* to be the most frequent etiology of CPI at IPF diagnosis.

NTM lung disease has been increasing over the past few decades in many areas of the world [19, 22, 23] including Japan [24]. The annual incidence has been reported as 2.7–5.6/100,000 in the United States [22, 25], 0.72–0.74/100,000 in France [26], 0.9–2.9/100,000 in the United Kingdom [27], 2.2–3.2/100,000 in Australia [28], 1.08/100,000 in Denmark [29], and 4.6–11.0/100,000 in Japan [30, 31]. In the present study, the incidence density of NTM lung disease and MAC lung disease was 9.71 and 8.04 cases per 1000 person-years, respectively. Thus, the incidence of NTM in patients with IPF would be higher than that in the general population.

Previous studies reported several risk factors for NTM infection such as gastroesophageal reflux disease, COPD, cystic fibrosis, history of TB and immunodeficiency or related diseases, and corticosteroid use [19]. Ide et al [31] reported that in 601 patients with NTM, interstitial lung disease was found in 24 (4.0%) patients. Hayashi et al [10] reported that in 634 patients with MAC lung disease, IPF was found in 28 (4.4%). Mirsaeidi et al [32] reported that

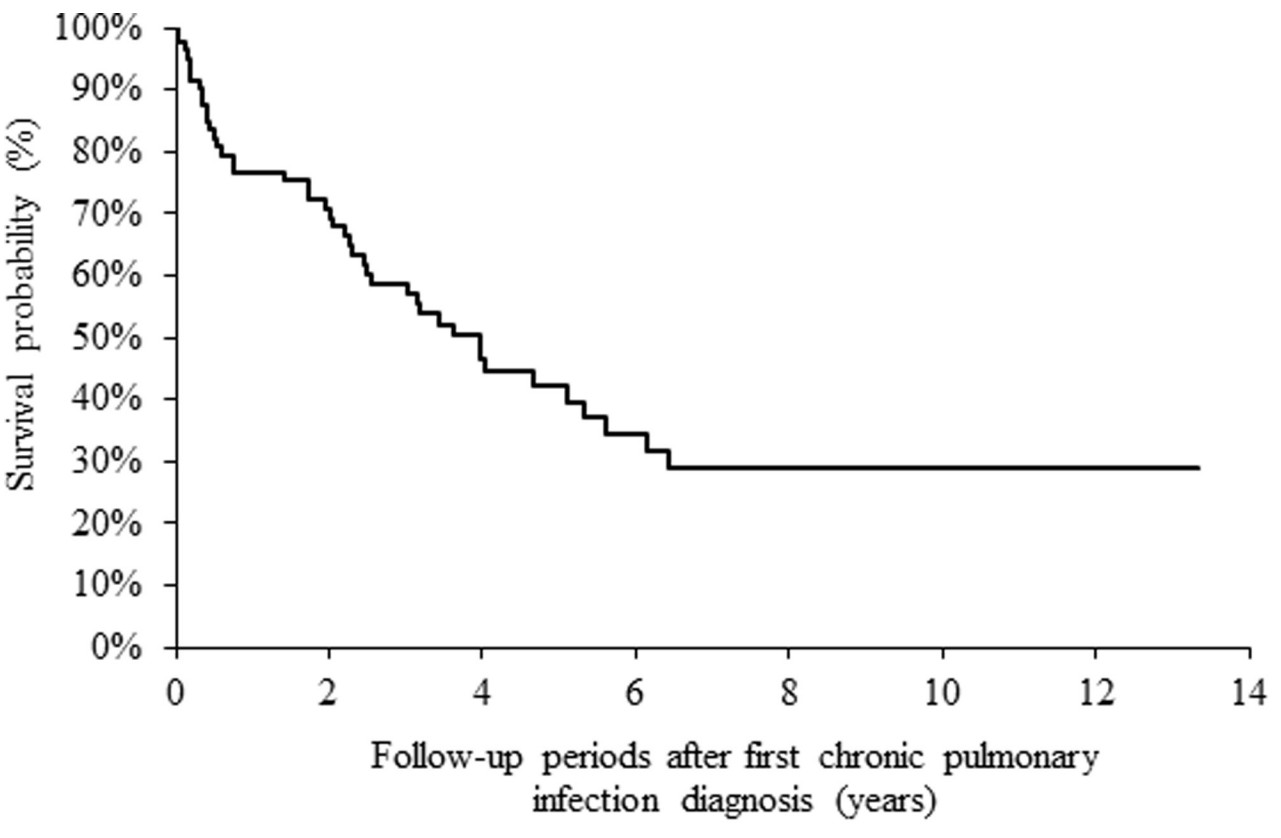

**Fig 4. Kaplan-Meier survival curves of all-cause mortality after CPI (including groups B, C, and D).**

compared to TB-related mortality, COPD, bronchiectasis, HIV, interstitial lung diseases, and tobacco use were significantly more common in persons with nontuberculous mycobacteria-

**Table 4. Causes of death in patients with idiopathic pulmonary fibrosis (IPF) with or without chronic pulmonary infections.**

| | With chronic pulmonary infections (N = 44) | Without chronic pulmonary infections (N = 254) |
|---|---|---|
| Cause of death | | |
| Progression of IPF | 10 (22.7) | 100 (38.6) |
| Acute exacerbation of IPF | 10 (22.7) | 42 (16.9) |
| Lung cancer | 3 (6.8) | 34 (13.4) |
| Pneumonia | 5 (11.4) | 21 (8.7) |
| Pneumothorax | 2 (4.5) | 4 (1.6) |
| Chronic pulmonary aspergillosis | 3 (6.8) | 0 |
| Pulmonary tuberculosis | 1 (2.3) | 0 |
| Pneumocystis pneumonia | 0 | 3 (1.2) |
| Microscopic polyangiitis | 0 | 2 (0.8) |
| Non-pulmonary | 10 (22.7) | 26 (10.2) |
| Unknown | 0 | 22 (8.7) |

related deaths. The results of these and the present study would indicate IPF to be a risk factor for NTM development.

CPA affects patients with no or only subtle generalized immune compromise but with an underlying lung condition [33–36]. Saraceno et al [37] reported that interstitial lung disease was found in 15 of 59 (25.4%) patients with CPA. In CPA case series, IPF was found in 2–11.9% of the patients [11–13]. In the present study, nearly equal numbers of patients developed CPA or MAC lung disease during follow-up, which would indicate IPF to be a risk factor for CPA development.

Genus *Nocardia* is a ubiquitous group of environmental bacteria that usually manifest as an opportunistic infection in immunocompromised hosts [38]. However, up to one-third of patients with nocardiosis are immunocompetent. Pulmonary nocardiosis is the most common clinical presentation of infection because inhalation is the primary route of bacterial exposure. The onset of symptoms may be subacute to more chronic. Kurahara et al [16] reported that of 59 patients with pulmonary nocardiosis, 55 (93%) had underlying pulmonary disease. Only one patient was being treated with corticosteroids, and no other patient was receiving immunosuppressive agents. Farina et al [14] reported that among 30 patients with nocardiosis, only one patient had lung fibrosis. Maeno et al [15] reported a patient who developed pulmonary nocardiosis during immunosuppressive therapy for IPF. These studies and the present study indicate that IPF would be one of the predisposing risk factors for the development of pulmonary nocardiosis.

As noted above, patients with altered lung structure have reduced local immunity and are more susceptible to infection [20]. CPI is likely to be secondary to the degree of structural destruction underlying the lung parenchyma found in IPF.

One important biomarker for IPF is the high-molecular-weight glycoprotein KL-6. An elevated serum KL-6 level ($\geq$1000 U/mL) in IPF patients at the initial visit was associated with increased mortality [39]. Baseline serum KL-6 level is also a sensitive predictor of the onset of acute exacerbation of IPF [40]. In a Kaplan-Meier analysis, patients with a baseline serum KL-6 level $\geq$1300 U/mL experienced earlier onset of acute exacerbation [40]. In the present study, a multivariate Cox proportional hazard model found a KL-6 level $\geq$2000 U/mL to be associated with CPI development. Increased serum KL-6 in IPF reflects impairment and the production of type II alveolar epithelial cells and is strongly correlated with fibrotic lesions [41, 42]. In addition, serum KL-6 levels are known to be high in infections such as pulmonary tuberculosis, and destruction of lung tissue due to infection is thought to be one of the factors [43]. Therefore, the prognosis of the patients with lung infection and structural changes (structural modification) in the lungs may have been worse. From the above two points, in IPF with CPI, KL-6 is elevated, and we consider that progression of these lung lesions can affect survival.

A multivariate Cox proportional hazard model also found PaO$_2$ <70 Torr to be associated with CPI development. Andréjak et al [44] assessed the risk factors of NTM development. Overall, they found chronic respiratory disease and chronic respiratory failure to be associated with a 16.5-fold (95% CI 12.2–22.2) and a 28.1-fold (95% CI 9.2–85.8) increase in the risk of NTM pulmonary disease, respectively. Kurosaki et al [45] reported that the results of their comparison of cases of interstitial pneumonia with and without pulmonary aspergillosis showed that the use of home oxygen therapy was significantly associated with pulmonary aspergillosis development, whereas the use of steroids and/or immunosuppressive agents was not. These studies and ours would indicate that respiratory failure would be a risk factor for CPI development.

One limitation of this study is that it was retrospective, so some clinical and laboratory findings were not available. Second, our conclusions are limited because this study was a single-center review. Third, the true cumulative incidence of CPI may be underestimated because

routine screening for the complications of CPI was lacking. Fourth, because we assessed the risk factors of CPI development that could be ascertained at IPF diagnosis, we could not elucidate whether corticosteroid and/or immunosuppressant therapy, which was usually administered after follow-up, would increase the risk of CPI development. Finally, our institution is a tertiary referral care for non-tuberculosis respiratory diseases and tuberculosis. Because our hospital has 51 beds for TB and TB patients tend to be referred to our hospital, it would be reasonable for TB to be the most frequent etiology of CPI at IPF diagnosis. Thus, there may be referral bias. However, no patient was referred for suspected NTM although referral bias from visits for cough or sputum workup cannot be completely ruled out. In the present study, the incidence density of NTM and MAC was 9.71 and 8.04 cases per 1000 person-years, respectively. Thus, the incidence of NTM in patients with IPF would be higher than that in the general population. There might be a bias toward TB, but the results of NTM suggest that patients with IPF have a greater risk of developing chronic infections than does the general population.

In conclusion, the present study showed that CPI was an important complication in patients with IPF. The etiologies of CPI were *M. tuberculosis*, NTM, and *Aspergillus* and *Nocardia* species. The incidence density of these CPIs would be much higher than that of the general population. Because a $PaO_2$ <70 Torr and KL-6 $\geq$2000 U/mL were found to be risk factors for CPI development, greater attention should be paid to IPF patients with these risk factors during follow-up. Further, because concurrent or sequential mixed infection was relatively common, if IPF patients with CPI deteriorate during antimicrobial therapy, the possibility of the development of another infection should be considered.

## Acknowledgments

We offer our sincerest thanks to Drs. Tsutomu Yanagisawa, Youtaro Takaku, and Kazuyoshi Kurashima of the Department of Respiratory Medicine, Saitama Cardiovascular and Respiratory Center, for their handling of the diagnosis and treatment of the patients with IPF.

## Author Contributions

**Investigation:** Kyuto Odashima, Naho Kagiyama, Tetsu Kanauchi.

**Methodology:** Kyuto Odashima, Noboru Takayanagi.

**Project administration:** Kyuto Odashima.

**Supervision:** Takashi Ishiguro, Noboru Takayanagi.

**Validation:** Takashi Ishiguro, Noboru Takayanagi.

**Writing – original draft:** Kyuto Odashima.

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
