## [Decision Letter · Decision Letter 0]

19 Dec 2019

PONE-D-19-31648

Incidence and etiology of chronic pulmonary infections in patients with idiopathic pulmonary fibrosis

PLOS ONE

Dear Dr. Odashima,

Thank you for submitting your manuscript to PLOS ONE. After careful consideration, we feel that it has merit but does not fully meet PLOS ONE’s publication criteria as it currently stands. Therefore, we invite you to submit a revised version of the manuscript that addresses the points raised during the review process.

We would appreciate receiving your revised manuscript by Feb 02 2020 11:59PM. To enhance the reproducibility of your results, we recommend that if applicable you deposit your laboratory protocols in protocols.io, where a protocol can be assigned its own identifier (DOI) such that it can be cited independently in the future. For instructions see: http://journals.plos.org/plosone/s/submission-guidelines#loc-laboratory-protocols

We look forward to receiving your revised manuscript.

Kind regards,

Vinicio A. de Jesus Perez, MD

Academic Editor

PLOS ONE

Journal Requirements:

1.

2. Please specify in your ethics statement whether participant consent was written or verbal. If verbal, please also specify: 1) whether the ethics committee approved the verbal consent procedure, 2) why written consent could not be obtained, and 3) how verbal consent was recorded.

3.  We noticed you have some minor occurrence of overlapping text with previous publications, which needs to be addressed. In your revision ensure you cite all your sources (including your own works), and quote or rephrase any duplicated text outside the methods section. Further consideration is dependent on these concerns being addressed. These are the references with which we found overlap:

https://openres.ersjournals.com/content/4/1/00111-2016

https://www.ncbi.nlm.nih.gov/pmc/articles/PMC3289849/

https://doi.org/10.1164/rccm.201807-1255ST

Reviewers' comments:

Reviewer's Responses to Questions

**Comments to the Author**

1. Is the manuscript technically sound, and do the data support the conclusions?

Reviewer #1: Yes

Reviewer #2: Partly

2. Has the statistical analysis been performed appropriately and rigorously? 

Reviewer #1: Yes

Reviewer #2: No

3. Have the authors made all data underlying the findings in their manuscript fully available?

Reviewer #1: Yes

Reviewer #2: No

4. Is the manuscript presented in an intelligible fashion and written in standard English?

Reviewer #1: Yes

Reviewer #2: Yes

5. Review Comments to the Author

Reviewer #1: Incidence and etiology of chronic pulmonary infections in patients with idiopathic pulmonary fibrosis

Authors : Odashima K et al.

This study is a retrospective study on the chronic pulmonary infections in patients with IPF. Study is very well done and has its own limitation due to the retrospective nature of it which authors have acknowledged in the manuscript.

I have few minor comments

“Five- and ten-year mortality rates of patients with and without CPI development were 43.4%

and 33.0%, and 68.9% and 72.8%, respectively.” In this statement in mortality section, are they trying to say that mortality is higher in those without infections? Although there was no difference in the KM curve. Please explain

The authors found KL 6 levels to be associated with survival in those with CPI. How do they explain this finding when KL6 level was elevated with diagnosis of IPF.

I would recommend to add a separate table on KL6 data findings. Showing how its different in each group and the trend in those who had CPI. This is an interesting finding of this paper as KL6 is not routinely used for treatment or evaluation of IPF world wide.

Regarding diagnosis of IPF, the authors have mentioned using ATS criteria. By which I am assuming all these patients had HRCT and a multidisciplinary evaluation. If yes, it would be informative to know how many of these were definite IPF or probable or possible and then showing the prevalence of these infection based on the type of IPF.

Reviewer #2: In this manuscript Odashima et al. describe the incidence and etiology of chronic pulmonary infections in patients with idiopathic pulmonary fibrosis. The authors define chronic pulmonary infection and as infection that was diagnosed 1 or more months after the onset of symptoms or upon the appearance of new shadows on pulmonary radiologic images, and note that nontuberculous mycobacteria, Mycobacterium tuberculosis, aspergillosis and nocardia were therefore most frequent etiologies of chronic pulmonary infection in patients with idiopathic pulmonary fibrosis.

The authors make a very good observation regarding chronic infections in patients with interstitial lung disease. This is been well described in patients with other chronic lung conditions such as chronic obstructive pulmonary disease, but much less so in the interstitial lung diseases. However, I have the following concerns and questions regarding the manuscript which need to be addressed.

The authors institution is a tertiary referral care for non-tuberculosis respiratory diseases and tuberculosis. Given that, it is probably reasonable to resume that it is also a referral center for complex patients with nontuberculous mycobacteria and other chronic infections. Given that older patients may be predisposed to these infections, and that the same population group is at a high risk for idiopathic pulmonary fibrosis as well, unless we can factor in the denominator, in addition to the expected baseline rate of idiopathic pulmonary fibrosis, nontuberculous mycobacteria, tuberculosis and other infections, one can question whether the association between idiopathic pulmonary fibrosis and chronic infections is spurious, and is secondary to a referral bias. For example, if the expected prevalence of idiopathic pulmonary fibrosis in a population is 10% and the expected prevalence of chronic infections is 5%, then just by chance alone, 0.5% of patients will have both the diseases, and if these patients are somehow referred to a single institution, simply analyzing these patients with both diseases can give a false impression that there is an association when there actually is not. This could be a difficult task to accomplish but the authors should try to address this in the manuscript.

The authors have divided the groups based on when IPF and CPI respectively were diagnosed in relationship to each other. Table 1 mentions steroids and immunosuppressants in these patients without alluding to the temporal relationship between the infection and immunosuppression. It probably makes more sense to divide the groups based on whether these patients had no risk factors, or had risk factors such as being immunosuppressed, or receiving corticosteroids before a diagnosis of CPI was made. These subgroups can be further divided based on the whether the IPF was diagnosed first, CPI, or at the same time.

Group C includes patients who had chronic infections at the time of diagnosis but none at the time of follow-up. Is this because these infections were treated? Or do they represent false positives?

Table 3. Multivariate Cox progression final model. E SR. It is unclear why there are multiple groups for this. Perhaps best to ignore those with unknown ESR and divide the rest into 2 groups only, less than 20, and more or equal to 20.

Figure 3. If the difference between these 2 groups is not significant, it is unclear how this figure adds to the manuscript.

Figure 4. Given the numerous factors affecting mortality in this group, this is likely not needed.

Table 4. To patients with microscopic polyangiitis were included in the group. Usual interstitial pneumonia pattern has been well described in patients with microscopic polyangiitis, and can in fact precede the diagnosis of microscopic polyangiitis. This patients do not have IPF and should not be included in the group.

Did all the patients with the organism isolates meet relevant clinical criteria for infection?

The authors should speculate on the reason for CPI in patients with IPF and hypothesize on possible mechanisms.

6. PLOS authors have the option to publish the peer review history of their article (what does this mean?). If published, this will include your full peer review and any attached files.

Reviewer #1: No

Reviewer #2: No

---

## [Author Response · Author response to Decision Letter 0]

7 Feb 2020

We made a rebuttal letter titled 'Response to Reviewers' that responds to each point raised by the academic editor and reviewers.

---

## [Decision Letter · Decision Letter 1]

9 Mar 2020

Incidence and etiology of chronic pulmonary infections in patients with idiopathic pulmonary fibrosis

PONE-D-19-31648R1

Dear Dr. Odashima,

We are pleased to inform you that your manuscript has been judged scientifically suitable for publication and will be formally accepted for publication once it complies with all outstanding technical requirements.

With kind regards,

Vinicio A. de Jesus Perez, MD

Academic Editor

PLOS ONE

Additional Editor Comments (optional):

Reviewers' comments:

Reviewer's Responses to Questions

**Comments to the Author**

1. If the authors have adequately addressed your comments raised in a previous round of review and you feel that this manuscript is now acceptable for publication, you may indicate that here to bypass the “Comments to the Author” section, enter your conflict of interest statement in the “Confidential to Editor” section, and submit your "Accept" recommendation.

Reviewer #1: All comments have been addressed

2. Is the manuscript technically sound, and do the data support the conclusions?

Reviewer #1: Yes

3. Has the statistical analysis been performed appropriately and rigorously? 

Reviewer #1: Yes

4. Have the authors made all data underlying the findings in their manuscript fully available?

Reviewer #1: Yes

5. Is the manuscript presented in an intelligible fashion and written in standard English?

Reviewer #1: Yes

6. Review Comments to the Author

Reviewer #1: The authors have addressed all the comments raised by me appropriately. Although, authors have not provided a separate table for KL6 in each group and I acknowledge their explanation but I feel its important to know in general. May be a table can be added as a supplementary data if not in main manuscript.

Apart from this I do not have any further comments.

7. PLOS authors have the option to publish the peer review history of their article (what does this mean?). If published, this will include your full peer review and any attached files.

Reviewer #1: Yes: Sandeep Sahay